# The Role of Nuclear Factor of Activated T Cells 5 in Hyperosmotic Stress-Exposed Human Lens Epithelial Cells

**DOI:** 10.3390/ijms22126296

**Published:** 2021-06-11

**Authors:** Gyu-Nam Kim, Young-Sool Hah, Hyemin Seong, Woong-Sun Yoo, Mee-Young Choi, Hee-Young Cho, Seung Pil Yun, Seong-Jae Kim

**Affiliations:** 1Department of Ophthalmology, Institute of Health Sciences, College of Medicine, Gyeongsang National University, Gyeongsang National University Hospital, Jinju 52727, Korea; sayuh@naver.com (G.-N.K.); seong_hm@daum.net (H.S.); oocee@daum.net (W.-S.Y.); 9985086@naver.com (M.-Y.C.); 2Biomedical Research Institute, Gyeongsang National University Hospital, Jinju 52727, Korea; yshah@gnu.ac.kr (Y.-S.H.); boiledbread@hanmail.net (H.-Y.C.); 3Department of Pharmacology and Convergence Medical Science, Institute of Health Sciences, College of Medicine, Gyeongsang National University, Jinju 52727, Korea

**Keywords:** human lens epithelial cell, hyperosmotic stress, necroptosis, NFAT5

## Abstract

We investigated the role of nuclear factor of activated T cells 5 (NFAT5) under hyperosmotic conditions in human lens epithelial cells (HLECs). Hyperosmotic stress decreased the viability of human lens epithelial B-3 cells and significantly increased NFAT5 expression. Hyperosmotic stress-induced cell death occurred to a greater extent in NFAT5-knockout (KO) cells than in NFAT5 wild-type (NFAT5 WT) cells. Bcl-2 and Bcl-xl expression was down-regulated in NFAT5 WT cells and NFAT5 KO cells under hyperosmotic stress. Pre-treatment with a necroptosis inhibitor (necrostatin-1) significantly blocked hyperosmotic stress-induced death of NFAT5 KO cells, but not of NFAT5 WT cells. The phosphorylation levels of receptor-interacting protein kinase 1 (RIP1) and RIP3, which indicate the occurrence of necroptosis, were up-regulated in NFAT5 KO cells, suggesting that death of these cells is predominantly related to the necroptosis pathway. This finding is the first to report that necroptosis occurs when lens epithelial cells are exposed to hyperosmolar conditions, and that NFAT5 is involved in this process.

## 1. Introduction

Cataracts are a major cause of blindness worldwide and affect people older than 50 years [1]. The performance of cataract surgery has increased significantly in recent years [2]. Understanding the pathophysiology of cataract formation is important for public health and for enhancing medical knowledge. Epidemiological studies demonstrated that sodium intake plays an important role in the pathogenesis of cataracts [3,4,5]. High-sodium intake caused cataracts in an animal model [6]. The plasma salt concentration, which depends on the balance between intake and excretion of salt, is the most important factor for determining plasma osmolality [7,8]. In addition, a deficit of total body water is common in elderly individuals [9,10]. The prevalence of dehydration among inpatients and long-term care residents has been reported to reach 80% [11].

Osmotic stress affects the kidneys, skin, respiratory organs, and eyes and is an important mechanism underlying tissue damage in these organs [12,13,14,15]. However, the cellular processes leading to cataract formation in lenses exposed to osmotic stress are not fully understood. High extracellular osmotic pressure causes water leakage and, thus, cell shrinkage, which is an early event in apoptotic cell death [16]. However, adaptive mechanisms can restore the osmotic balance and enable cells to survive upon exposure to osmotic stress. The classical cellular response to high extracellular osmolality involves the transcription of target genes by nuclear factor of activated T cells 5 (NFAT5), also known as tonicity-responsive enhancer-binding protein (TonEBP/OREBP), a member of the Rel family of transcription factors [17]. NFAT5 is activated by phosphorylation in a hyperosmotic environment [18]. The target genes of NFAT5 include those encoding enzymes and transporters that are implicated in intracellular accumulation of organic osmolytes such as sorbitol, myo-inositol, betaine, and taurine [17,19].

The mechanisms responsible for osmoadaptation to hyperosmolarity in renal cells are well established [20,21]. Activation of NFAT5 and subsequent transactivation of osmoprotective genes allow osmolytes to accumulate in the cytoplasm, which reduces the intracellular ionic strength [20,21]. In addition, NFAT5 was recently shown to be involved in the mechanisms responsible for osmoadaptation to hyperosmotic stress in human retinal pigment epithelial cells and limbal epithelial cells [22,23]. Recent studies suggested that cataract formation is initiated in the lens epithelium [24,25,26,27]. Therefore, the role of lens epithelial cells in defense against hyperosmotic stress and the mechanisms by which damage/injuries occur must be investigated. Only a few reports have evaluated the expression and function of NFAT5 in human lens epithelial cells (HLECs) [28]. This study investigated whether NFAT5 is activated by hyperosmotic stress in cultured HLECs and determined its role in these cells.

## 2. Results

### 2.1. Effect of Hyperosmotic Stress on the Viability of Human Lens Epithelial B-3 (HLE-B3) Cells

Morphological changes were investigated by phase-contrast microscopy (Figure 1A). HLE-B3 cells exhibited marked rounding and eventually detached from the culture dish as the culture duration and osmolarity increased. They also displayed retraction and plasma membrane blebbing, which are typical morphological features of apoptosis. In summary, progressive morphological changes occurred in HLE-B3 cells as osmolarity increased. Furthermore, hyperosmotic stress induced decrease of the viability of HLE-B3 cells in a dose-dependent manner in comparison with control cells (Figure 1B).

### 2.2. Effect of Hyperosmotic Stress on Expression of NFAT5 in HLE-B3 Cells

It was confirmed that the *NFAT5* mRNA level significantly increased when 100 or 150 mOsm/L NaCl were added to normal media for 24 h, respectively (*p* = 0.0058, *p* < 0.0001, respectively) (Figure 2A). In addition, NFAT5 protein expression level increased by 2.4 and 3.2 times, respectively, compared to NM when exposed to 100 and 150 mOsm/L NaCl for 24 h (*p* = 0.0445, *p* = 0.0286, respectively, Figure 2B,C). The hTonE site is a specific binding site for *NFAT5* and has been used to monitor the induction of NFAT5 expression in response to hyperosmotic stimuli [29]. To measure the transcriptional activity of the endogenous NFAT5 protein, HLE-B3 cells were transiently transfected with the hTonE-GL3 reporter construct and stimulated with hyperosmotic stress. The hTonE reporter was significantly activated upon stimulation with sodium chloride in a dose-dependent manner (Figure 2D).

### 2.3. NFAT5 Knockout Cell Production and Changes of Cell Viability after Hyperosmotic Stress in HLE-B3 Cells

To determine the role of NFAT5 in HLECs exposed to hyperosmotic stress, we used clustered, regularly interspaced, short, palindromic repeats (CRISPR) RNA-guided Cas9 nucleases to knockout *NFAT5* in HLE-B3 cells. Sanger sequencing of the targeted regions is shown in Figure 3A. We first established cell lines constitutively expressing the Cas9 nuclease (HLE-B3-Cas9). HLE-B3-Cas9 cells were transiently transfected with a single guide RNA (5′-ACAAACACTTGCAACACTA C-3′) targeting the seventh exon of the *NFAT5* gene and then seeded as single cells into 96-well plates for clonal expansion (Figure 3B). Some clones showed the complete absence of NFAT5 expression (NFAT5 KO) compared with wild-type control cells (NFAT5 WT) based on Western blot analysis (Figure 3C). To verify the absence of NFAT5, we performed the luciferase reporter assay. The transcriptional function of NFAT5 was abolished in NFAT5 KO cells (Figure 3D,E). Next, we examined changes of cell viability in response to hyperosmotic stress. Morphological changes were investigated by phase-contrast microscopy (Figure 3F,H). NFAT5 KO cells exhibited marked rounding as the culture duration and osmolarity increased and eventually detached from the culture dish. Moreover, hyperosmotic stress (+150 mOsm/L NaCl during 24 h and 48 h) induced cell death to a greater extent in NFAT5 KO cells than in NFAT5 WT cells (Figure 3G,I).

### 2.4. Effect of Hyperosmotic Stress on the Cell Cycle in HLE-B3 Cells

The percentage of cells in S phase significantly decreased upon culture in medium with an osmolarity of 150 mOsm/L NaCl for 24 h in the NFAT5 WT and KO cells, but did not decrease for 48 h (Figure 4A,C). Meanwhile, the portion of cells in sub-G1 phase significantly increased upon culture in medium with an osmolarity of 150 mOsm/L NaCl for 24 and 48 h in both groups (Figure 4B,D). Particularly, sub-G1 portions in NFAT5 KO cells increased more than about two times in 24 and 48 h compared to WT cells (Figure 4B,D).

### 2.5. Characterization of Hyperosmotic Stress-Induced Death of Lens Epithelial Cells

The role of NFAT5 on hyperosmotic stress-induced death of HLE-B3 cells was investigated using the NFAT5 KO and WT cell lines. Viabilities of both cell lines exposed to hyperosmotic stress were significantly lower than that of non-stressed control cells. As expected, NFAT5 KO cells were more sensitive to hyperosmotic stress than NFAT5 WT cells (Figure 5A). Pre-treatment with z-VAD-fmk, a caspase inhibitor, significantly attenuated the hyperosmotic stress-mediated reduction in viability of NFAT5 WT cells and NFAT5 KO cells, but there was no difference in the effect of z-VAD treatment between two groups (Figure 5A). Next, to explore the mechanism by which hyperosmotic stress induces apoptosis of NFAT5 WT cells, we examined endogenous expression of Bcl-2 family members, including Bax, Bcl-2, and Bcl-xl. Quantitative PCR and Western blot analyses showed that anti-apoptotic Bcl-2 and Bcl-xl expressions after hyperosmolar stress (+150 mOsm/L NaCl) during 48 h were reduced in NFAT5 WT and NFAT5 KO cells (Figure 5B–F). However, these changes in pro-apoptotic and anti-apoptotic markers between the two groups were not statistically significant (Figure 5B–F).

Then, we investigated the effects of necrostatin-1 (NEC), a specific inhibitor of necroptosis, on hyperosmotic stress-induced death of NFAT5 KO and WT cells. Pre-treatment with NEC significantly blocked hyperosmotic stress-induced death of NFAT5 KO cells, but not of NFAT5 WT cells (Figure 6A). The initiation of necroptosis requires receptor-interacting protein kinase 1 (RIP1) and RIP3 [30]. RIP1 and RIP3 induce cross-phosphorylation of the Ser166 and Ser227 residues, respectively [31]. Therefore, the phosphorylation state of RIP1 and RIP3 is closely related to protein kinase activity. We detected the expression and phosphorylation of these proteins to verify the occurrence of necroptosis by Western blot analysis. The phosphorylation levels of RIP1 (Ser166) and RIP3 (Ser227) were up-regulated in NFAT5 KO cells compared to WT cells (Figure 6C,D).

### 2.6. Changes in Inflammatory Cytokines by qPCR after Exposure to Hypertonic Conditions

We measured changes in inflammatory cytokines by qPCR after 24 h exposure to hypertonic condition in NFAT5 WT and KO cell lines (Figure 7). When exposed to hyperosmolar stress in WT cells, no statistically significant changes in TNF-α, IL-1α, IL-1β, and IL-6 were observed. In addition, compared to WT cells, mRNA levels of TNF-α, IL-1β, and IL-6 were increased in normal media in KO cells, so it could be seen that NFAT5 plays a certain role in the induction of inflammation. Additionally, IL-1α and IL-1β were found to increase when exposed to hypertonic condition in the NFAT5 KO cell lines.

## 3. Discussion

This current study was investigated about the cell death under hyperosmolar stress in HLECs. Exposure to hyperosmolar stresses reduces HLEC viability and causes cell cycle arrest. Programmed cell death plays an important role in response to harmful stress and the regulation of homeostasis and various diseases. Apart from the classical apoptosis, recent basic experimental studies also discovered novel programmed cell death, necroptosis. This is the first programmed cell death that does not depend on the activation of caspases. Especially, RIP1, RIP3, and MLKL are the main molecules involved in necroptosis. In this study, the decrease in cell viability was not recovered when z-VAD, a caspase inhibitor that has a role in blocking apoptosis, was treated to NFAT5-WT and NFAT5-KO cells. On the other hand, when the RIP1 kinase inhibitor, Nec-1, was treated, especially after hyperosmolar exposure in NFAT5-KO cells, it was confirmed that the decrease in cell viability was recovered. In addition, reductions in phopsporylated-RIP1 and -RIP3 also were identified. Therefore, our results suggest that the osmo-adaptive response through NFAT5 enhances HLEC viability by suppressing necroptosis. To the best of our knowledge, this study is the first to describe the role of NFAT5 in HLECs upon hyperosmotic stress.

The lens lacks blood flow, and fluid circulation caused by the center-to-surface hydrostatic pressure gradient is a functional replacement for blood flow [32]. Therefore, osmotic homeostasis in the lens is thought to play a very important role in maintaining the lens environment. The lenticular sodium content and cataract formation are increased in rats fed a high-sodium diet [6]. These findings suggest that sodium in the systemic circulation efficiently passes through the blood–aqueous humor barrier and reaches the lens. However, it is unknown whether a high-salt diet affects the sodium content of the aqueous humor and lens in humans, and further research is needed to clarify this.

Apoptosis and necroptosis, the two major modes of programmed cell death, induce distinct biomolecular changes. In this study, hyperosmotic stress-exposed NFAT5 WT HLECs exhibited biomolecular changes consistent with apoptosis (i.e., down-regulation of anti-apoptotic proteins), whereas hyperosmotic stress-exposed NFAT5 KO HLECs displayed biomolecular changes consistent with necroptosis (i.e., activation of RIP1 and RIP3). Various factors, such as genetic factors, aging, radiation, metabolic factors, and other toxic substances, can induce cataract formation in vivo and in vitro [33]. However, a common cellular basis for cataract formation is not fully understood. Lens epithelial cell apoptosis is considered as common molecular basis of the initiation and subsequent formation of several types of cataracts. Elevated levels of oxygen radicals from H_2_O_2_ are reported in the aqueous humor of age-related cataract patients. Understanding of this mechanism of lens epithelial cell apoptosis may be key for opening the door to new treatment strategies [34]. In our study, as a result of measuring and using DCF-DA, it was confirmed that intracellular ROS production increased when exposed to hyperosmolar stress for 24 h in NFAT5 WT lens epithelial cells. However, in the NFAT5 KO cell lines, ROS production was less than in the WT cell line under hypertonic conditions (Appendix A). These results suggest that NFAT5 will play a role in ROS generation in lens epithelial cells. Many previous studies have shown that various pathways or genes play an important role in lens epithelial cell apoptosis. Although previous studies have shown that some genes, such as bcl-2, caspase 2, and hGSTA, play a main role in lens epithelial cell apoptosis, much work is still needed to elucidate these mechanisms of lens epithelial cell apoptosis [34,35,36,37,38]. It was recently reported that culture in hyperosmotic medium induces apoptosis of ocular cells [36,37,38]. These findings suggest that hypertonic stress also induces apoptosis of lens epithelial cells. Hyperosmotic stress may be initiated in various pathological conditions, which, in turn, leads to tissue damage. In this process, activation of survival signals with the ability to restore cell homeostasis determines cell fate.

Necroptosis is responsible for regulated necrosis and is induced by various pathological stimuli, such as tumor necrosis factor α (TNF- α), interferon γ, lipopolysaccharide, and pathogen- and damage-associated molecular patterns [39]. Recently, the role of necroptosis in the pathophysiology of several ocular diseases has been reported, in particular, necroptosis involved in degeneration of corneal epithelial and stromal cells [40,41], retinal ganglion cells, retinal pigment epithelial cells, or photoreceptors, and eventually develops corneal toxicity of eye drops, glaucoma, Leber congenital amaurosis, age-related macular degeneration, and retinitis pigmentosa [42]. However, to the best of our knowledge, there have been no studies about the role of necroptosis in the lens epithelial cell during cataractogenesis. In addition, no studies have been conducted on whether hyperosmolar stress in any kind of cells can induce necroptosis. The present study is the first to report that necroptosis occurs when lens epithelial cells are exposed to hyperosmolar conditions and that NFAT5 is involved in this process. NFAT5, a transcription factor sensitive to osmotic pressure, mediates the expression of genes related to cell survival under hyperosmotic conditions. The present study investigated whether complete depletion of NFAT5 regulated hyperosmotic stress-induced cell death. Previously, there were studies that increased TNF-α levels had an effect on necroptosis [39], but, in this study, TNF-α decreased in KO cells under hypertonic conditions compared to WT cells. On the other hand, it is thought that increased IL-1α and IL-1β expressions in the NFAT5 KO cell lines played a role in the necroptosis of HLECs under hyperosmotic stress. Hyperosmotic stress increased NFAT5 expression, which, in turn, triggered apoptosis through decreased expression of anti-apoptotic proteins (i.e., Bcl2 and Bcl-xL). By contrast, silencing of NFAT5 enhanced phosphorylation of RIP1 and RIP3, which are components of the necrosome. Therefore, the balance between NFAT5-mediated apoptosis and necroptosis may be critical to determine the fate of lens epithelial cells under hyperosmotic stress.

The major limitation of this study is that there was no mechanistic study on how depletion of NFAT5 is involved in necroptosis, that is, activation of RIP1 and RIP3. This mechanistic study of NFAT5 and necroptosis is a big project, and the authors are in progress on follow-up studies. The mechanism by which NFAT5 may be involved in necroptosis is as follows. First, it acts on changes in the expression of down-signaling pathways such as nuclear factor kappa-B (NF-κB). In other words, NFAT5 increases the activity of NF-κB, and NFAT depletion suppresses the activation of NF-κB and may be involved in necroptosis [43,44]. Second, to increase intracellular osmolyte, NFAT5 affects the expression of enzymes, such as aldose reductase, which are typically activated by NFAT5. It is thought that the depletion of NFAT5 reduces the activity of aldose reductases, and, as a result, necroptosis may be induced by several intermediate products [45].

The present study demonstrated that hyperosmotic stress significantly decreased the viability of HLECs in a dose- and time-dependent manner. These results are consistent with observations in corneal limbal epithelial cells [36]. In both cell types, NFAT5 was induced by hyperosmotic stress and cell viability was decreased when NFAT5 was suppressed. In addition to apoptosis, our results also indicate the involvement of necroptosis of HLECs. Given that necroptosis was more predominant than apoptosis in NFAT5 KO cells, NFAT5 is assumed to be related to the necroptosis pathway. Further investigation of the molecular mechanisms involved in necroptosis of HLECs may help to develop therapeutic strategies that can prevent and/or delay cataract development in humans and animals.

## 4. Materials and Methods

### 4.1. Cell Culture and Treatment

HLE-B3 cells, a HLEC line immortalized by viral transformation of SV-40, were purchased from the American Type Culture Collection (Manassas, VA, USA). These cells were maintained in Minimum Essential Medium (MEM; Cat # 11095080; Lot No. 2120598; Gibco, USA) containing 10% fetal bovine serum and 1% antimicrobial solution (Sigma-Aldrich, St. Louis, MO, USA) in 5% CO_2_ at 37 °C. In the entire experiment, when only media were used, they were described as normal media (NM), and hyperosmolar stress was applied by adding 50, 100, and 150 mOsm/L of NaCl to this baseline media. The osmolarity of the media itself could have a very important effect on this study, so we measured the osmolarity of the normal media using a 2430 Multi-Osmette auto-sampling turntable osmometer (Precision System Inc., Natick, MA, USA). As a result, the osmolarity of the normal media was measured as 291.5 ± 1.0 mOsm/L (*n* = 4). All cultures were grown to ~80–90% confluency prior to experiments. To induce hyperosmotic stress, sterile sodium chloride (1 M) was added to the culture media. Cells were incubated in hyperosmolar medium for 24, 48, or 72 h. An osmolarity range of 300–450 mOsm/L was selected based on previous data indicating that osmolarity in areas of tear breakup can reach up to 560 mOsm/L in the precorneal tear layer [26].

### 4.2. Measurement of Cell Viability

HLE-B3 cells were seeded into 24-well plates at a density of 5 × 10^4^ cells per well and incubated for 24 h. The following day, cells were cultured in medium with various osmolarities (NM, +50, +100, and +150 mOsm/L) for several time points (24, 48, and 72 h). Cell viability was measured using a CCK-8 (Dojindo Laboratories, Kumamoto, Japan). Briefly, 10 μL of CCK-8 solution were added to each well and incubated for 1 h at 37 °C in a humidified atmosphere containing 5% CO_2_. The amount of formazan dye generated by cellular dehydrogenases was determined by measuring absorbance at 450 nm using a microplate reader (Molecular Devices, Sunnyvale, CA, USA). Cells were observed using an inverted microscope (Nikon, Tokyo, Japan) to detect any phenotypic differences between treated and control cells.

### 4.3. Flow Cytometric Analysis

Cells were washed twice with cold phosphate-buffered saline (PBS), fixed with 70% ethanol for 1 h at 4 °C, treated with 1 mg/mL RNase A (Sigma-Aldrich, St. Louis, MO, USA), and stained with 50 μg/mL PI (Sigma-Aldrich, St. Louis, MO, USA). The relative DNA content per cell was measured using a flow cytometer (model FC500; Beckman Coulter, Brea, CA, USA). Data were analyzed using Beckman Coulter Cytomics CXP software (Applied Cytometry Systems, Dinnington, UK). Results were expressed as percentages of the total numbers of gated cells.

### 4.4. Transfection and Luciferase Assays

HLE-B3 cells were transfected with the hTonE-GL3 reporter construct (hTonE-GL3, a gift from Dr. S. N. Ho, University of California at San Diego, La Jolla, CA, USA) [40] using Lipofectamine 3000 (Life Technologies, Carlsbad, CA, USA). The hTonE-GL3 construct (0.5 μg) was co-transfected with 10 ng of pRL-CMV (Renilla luciferase). After transfection, cells were maintained in isotonic medium for 24 h and then switched to hypertonic medium or maintained in isotonic medium for another 24 h. Cells were washed twice with PBS and lysed with passive lysis buffer (Promega, Madison, WI, USA). Firefly and Renilla luciferase activities in extracts were measured using the Dual-Luciferase Reporter Assay System (Promega, Maddison, WI, USA) according to the manufacturer’s instructions. For each sample, Firefly luciferase activity was divided by Renilla luciferase activity to control for variation in the transfection efficiency.

### 4.5. Generation of the NFAT5 KO Cell Line

HLE-B3 cells were transfected with the pRGEN_Cas9_PuroR_RFP_CMV plasmid, which harbors the Cas9-red fluorescent protein (RFP) expression cassette (Toolgen, Inc., Seoul, Korea), and transfected cells were selected using puromycin (1 μg/mL). After 2 weeks of selection, each clone was transferred to a 24-well plate using a cloning cylinder (Fisher Scientific, Pittsburgh, PA, USA). Independent clones were isolated, and RFP expression was examined by fluorescence microscopy. In subsequent passages, cells were maintained in growth medium containing antibiotics. A 20-bp guide RNA sequence (5′-ACAAACACTTGCAACACTAC-3′) targeting DNA in the seventh exon of *NFAT5* was selected using Knockout Guide Design (Synthego, Redwood City, CA, USA; design.synthego.com). A total of 1.5 × 10^5^ clonally derived Cas9-expressing HLE-B3 cells was seeded into a 6-cm tissue culture dish and cultured until they reached 70–80% confluency. Cells were transfected with 30 µM of synthetic NFAT5-targeting single guide RNA, which was a fusion of crRNA and tracrRNA (Synthego, Menlo Park, CA, USA). Clonal cell lines with modification of the *NFAT5* gene were obtained by isolating single cells via serial dilution followed by expansion to obtain individual clones. Individual clones were lysed in RIPA lysis buffer (Santa Cruz Biotechnology, Santa Cruz, CA, USA) for 20 min at 4 °C. The lysate was centrifuged at 16,000 g for 10 min, and 20 μg of the supernatant was fractionated on 7.5% SDS-PAGE gels, transferred to nitrocellulose membranes, and immunoblotted with an anti-NFAT5 antibody. Genomic DNA was isolated from edited clones and non-edited control cells. Exon 7 of *NFAT5* was PCR-amplified using NFAT5-specific PCR primers (forward, CAGGAATTTAGCCAACTCTTTTAT; reverse, GCCAGTGTCATGTTGTTGCT). PCR products were cloned into pGEM-T Easy (Promega, Madison, WI, USA). Individually cloned amplicons were analyzed by Sanger sequencing (Bioneer, Seoul, Korea). To exclude the off-target effect by Cas9 knockout, cells were transfected with siRNA by reverse transfection against NFAT5 or AccuTarget Negative Control siRNA (Bioneer, Seoul, Korea), used as negative control siRNA. Briefly, siRNA (100 nm final concentration) and 1.5 μL Lipofectamine^®^ RNAiMAX in a total volume of 100 μL serum-free medium were prepared following the protocol of the manufacture and added to each well (6-well plate) and incubated at 37 °C 5% CO_2_ for 6 h. Then, siRNA was removed, and cells were incubated at 37 °C 5% CO_2_ for 24 h in normal growth medium (Appendix A). To summarize the results briefly, there was no effect on the mRNA changes up to NFAT-1, -2, and -3 in both cas9-knockout cells and siRNA-knockdown cells, and changes in TNF-α and IL-1β also showed similar results in both cells. The sequences of the siRNA constructed used for siRNA transfection are presented in Table 1.

### 4.6. Quantitative Real-Time PCR

Total RNA was extracted from cells at the indicated time points, and first-strand cDNA was synthesized using random hexamer primers provided in the first-strand cDNA synthesis kit (Applied Biosystems, Framingham, MA, USA) according to the manufacturer’s instructions. All primers and probes (NFAT5, Cat # Hs00232437 and GAPDH, Cat # Hs02758991) were commercially obtained (TaqMan^®^ Gene Expression Assay, Applied Biosystems, Waltham, MA, USA). Total RNA (1 μg) was used for cDNA synthesis on an iCycler thermocycler (Bio-Rad Laboratories, Hercules, CA, USA). The qPCR was performed using iQ Sybr Green supermix kit (Bio-Rad Laboratories, Hercules, CA, USA) by a Light Cycler 480 II (Roche Life Science, Indianapolis, IN, USA). PCR Primers were synthesized on the basis of reported cDNA sequences in the NCBI data bank. Sequences of the primers used for PCR were demonstrated following the set of primers (sense and antisense, respectively) in Table 2.

### 4.7. Western Blot Analysis

Cells were lysed in RIPA lysis buffer supplemented with Halt phosphatase inhibitor cocktail (Thermo Fisher Scientific, Waltham, MA, USA) and Halt protease inhibitor cocktail (Thermo Fisher Scientific, Waltham, MA, USA), sonicated, and centrifuged at 12,000× *g* for 10 min at 4 °C to remove insoluble debris. Protein concentrations of the cell lysates were determined using a BCA Protein Assay Kit (Pierce, Rockford, IL, USA). Whole cell lysates were separated by SDS-PAGE in a 10% polyacrylamide gel and transferred to a nitrocellulose membrane (Millipore, Bedford, MA, USA). After blocking with 5% nonfat dry milk, each blot was incubated with primary antibodies against Bax (sc-493, Santa Cruz Biotechnology, Inc., Santa Cruz, CA, USA), Bcl-xL (sc-634, Santa Cruz Biotechnology, Inc, Santa Cruz, CA, USA), Bcl-2 (sc-492, Santa Cruz Biotechnology, Inc, Santa Cruz, CA, USA), RIP1 (Cell Signaling Technology, Inc., Beverly, MA, USA), phospho-RIP1 (Cell Signaling Technology, Inc., Danvers, MA, USA), RIP3 (Abcam), phospho-RIP3 (Cell Signaling Technology, Inc., Danvers, MA, USA), and β-actin (Sigma-Aldrich), and then with horseradish peroxidase-conjugated anti-rabbit immunoglobulin (Ig) G or anti-mouse IgG (Cell Signaling Technology, Inc., Danvers, MA, USA). Antibody binding was detected using SuperSignal Chemiluminescent Substrate (Thermo Fisher Scientific, Waltham, MA, USA). Images were acquired using the ChemiDoc Touch Imaging System (Bio-Rad, Hercules, CA, USA). Densitometry was performed using ImageJ software (NIH, Bethesda, MD, USA).

### 4.8. DCFDA/H2DCFDA Cellular ROS Assay

Cells were plated on 96-well cell culture plates at 2.5 × 10^5^ cells/well, and cells were allowed to adhere overnight. The assay was performed following the protocol of DCFDA/H2DCFDA cellular ROS assay kit (ab113851; Abcam, Cambridge, UK). After staining, plate was detected immediately on a fluorescence plate reader (Tecan Systems, Inc., San Jose, CA, USA) at Ex/Em = 485/535 nm in end point mode in the presence of media.

### 4.9. Statistical Analysis

All data were analyzed using GraphPad Prism 6 software (San Diego, CA, USA). Quantifiable results are presented as the mean ± S.E.M of at least three independent experiments. Statistical significance was determined by an unpaired two-tailed Student’s test or a one-way ANOVA test followed by Tukey’s multiple comparisons test. Assessments with *p* < 0.05 were considered significant.

## 5. Conclusions

The present study indicates that hyperosmotic stress induces activation of NFAT5 in HLECs and that the osmo-adaptive response through NFAT5 enhances HLEC viability by suppressing necroptosis. Further research is required to elucidate the other functions of NFAT5 in HLECs and the signaling pathways that lead to cataractogenesis.

## Figures and Tables

**Figure 1 ijms-22-06296-f001:**
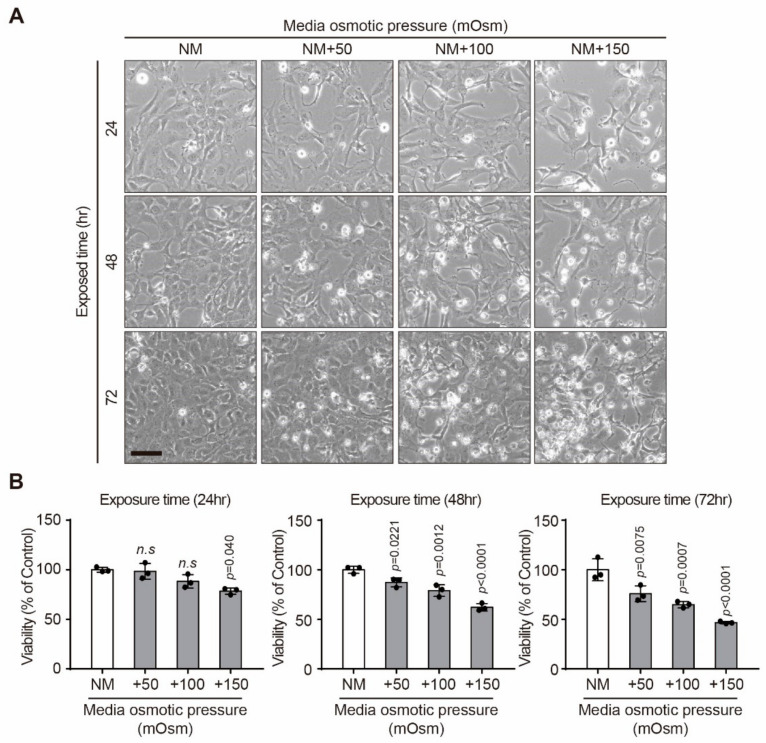
Effect of hyperosmotic stress on the viability of HLE-B3 cells. (**A**) HLE-B3 cells were cultured in medium with various osmolarities (normal media (NM), +50, +100, and +150 mOsm/L of NaCl) for several time points (24, 48, and 72 h). Representative phase-contrast images were acquired (scale bar = 50 µm). (**B**) The effect of hyperosmotic stress on the activity of mitochondrial dehydrogenases in HLE-B3 cells was measured by the CCK-8 assay. The viability of control (normal media, NM) cells was set to 100%. Viability as a percentage of that of control cells is shown. Bars represent the mean ± S.E.M. (*n* = 3); n.s; non-significance vs. NM, *p* < 0.05; significance vs. NM.

**Figure 2 ijms-22-06296-f002:**
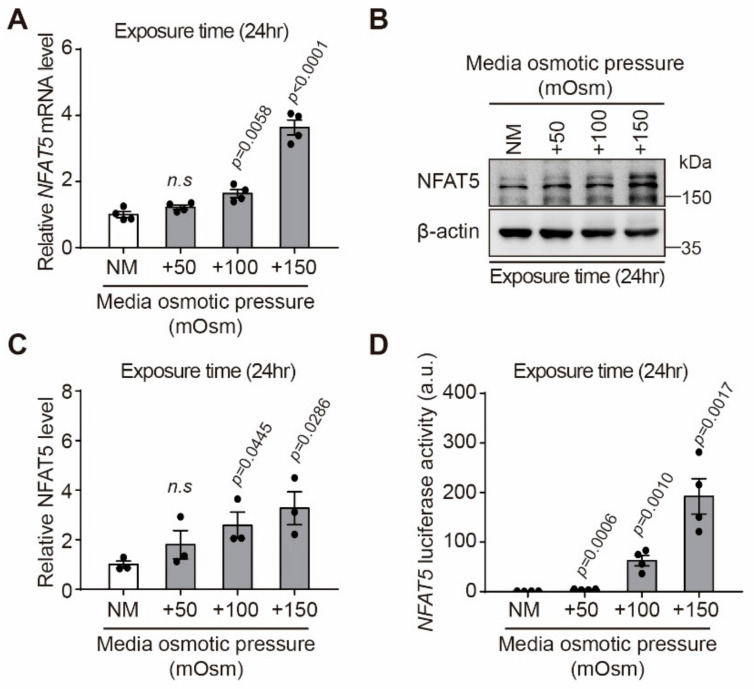
Effect of hyperosmotic stress on NFAT5 expression in HLE-B3 cells. HLE-B3 cells were cultured in medium with various osmolarities (NM, +50, +100, and +150 mOsm/L) for 24 h. (**A**) Hyperosmotic stress-induced NFAT5 mRNA expression (*n* = 4). (**B**) Representative Western blot image of expression level of NFAT5 after exposure to various concentrations of hyperosmolar stress. (**C**) Relative quantification graph of protein expression levels of NFAT5 (*n* = 3). (**D**) Relative luciferase activity in HLE-B3 cells cultured in NM and hypertonic culture medium (*n* = 4). Bars represent the mean ± S.E.M. n.s; non-significance vs. NM, Significance, *p* < 0.05 vs. NM; NM, normal media.

**Figure 3 ijms-22-06296-f003:**
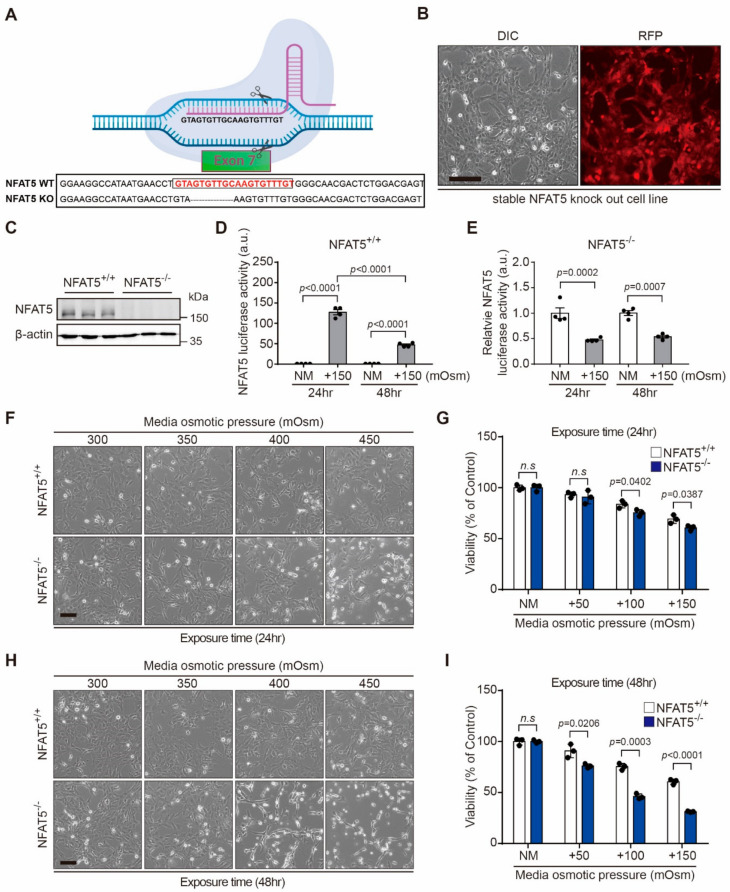
Deletion of NFAT5 increased susceptibility of cell death by hyperosmolar stress. (**A**) Sequence alignment of a WT allele from control HLE-B3-Cas9 cells and a mutant allele identified by genomic PCR. (**B**) The stable HLE-B3-Cas9 cell line. HLE-B3 cells were transfected with the pRGEN-Cas9-PuroR-RFP-CMV plasmid, and cells stably expressing Cas9 were selected with puromycin. Stable expression of red fluorescent protein (RFP) in the HLE-B3 cell line is shown (scale bar = 50 µm). (**C**) A representative Western blot showing that exposure to hyperosmotic stress (+150 mOsm/L) increased NFAT5 expression in NFAT5 WT cells, while NFAT5 was not detected in NFAT5 KO cells. The β-actin was used as a loading control. (**D**,**E**) Transcriptional activity of NFAT5 in NFAT5 KO and WT cells in response to hyperosmotic stress (+150 mOsm/L) (*n* = 3). (**F**) Representative phase-contrast images after hyperosmolar stress during 24 h were acquired (scale bar = 50 µm). (**G**) Comparison of the viability of NFAT5 KO and WT cells in response to various levels of osmotic stress during 24 h (*n* = 3). (**H**) Representative phase-contrast images were acquired after 48 h of hyperosmolar treatment (scale bar = 50 µm). (**I**) Comparison of the viability of NFAT5 KO and WT cells in response to various levels of osmotic stress during 48 h. The viability of untreated control cells was set at 100% (*n* = 3). Bars represent the mean ± S.E.M. n.s; non-significance, NM, normal media.

**Figure 4 ijms-22-06296-f004:**
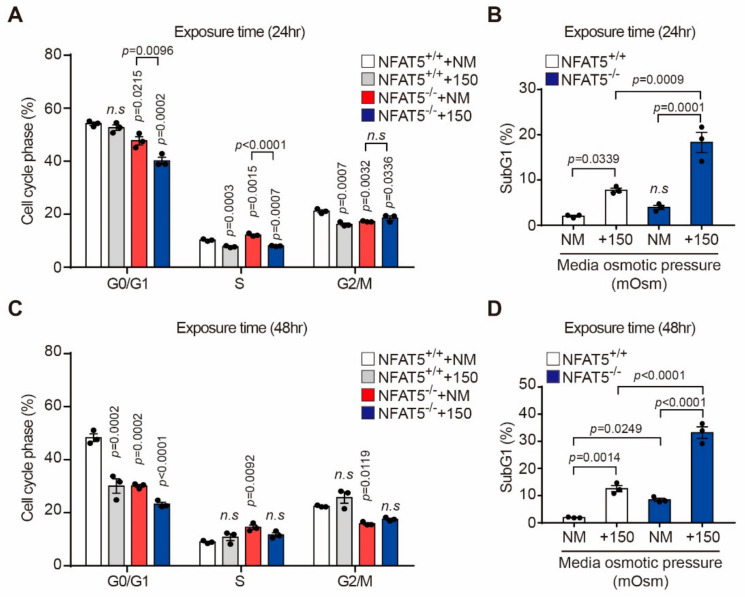
Effect of hyperosmotic stress on the cell cycle in HLE-B3 cells. HLE-B3 cells were cultured in normal medium with adding 150 mOsm/L NaCl for 24 and 48 h, and then analyzed by flow cytometry. The DNA content percentages of cells in various cell cycle phases (**A**,**C**) and the percentage of cells in sub-G1 phase (**B**,**D**) were determined (*n* = 3). Bars represent the mean ± S.E.M. n.s; non-significance vs. NM, NM, normal media.

**Figure 5 ijms-22-06296-f005:**
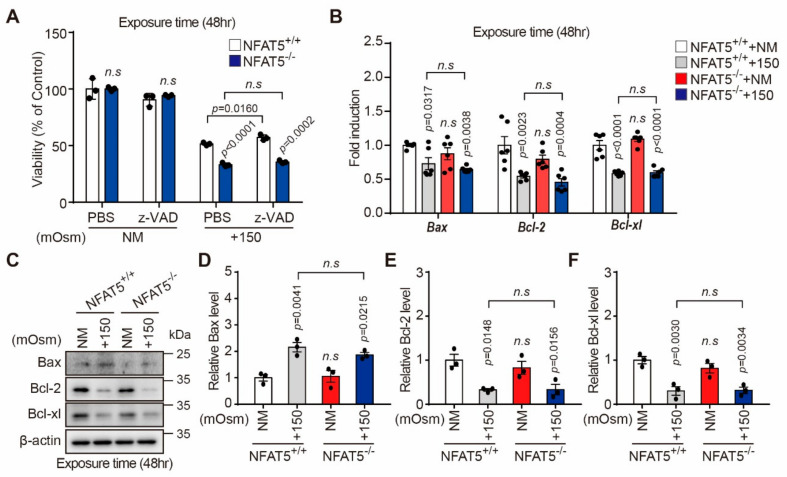
Effect of NFAT5 on hyperosmotic stress-induced cell death. (**A**) NFAT5 KO and WT cells were pre-treated with z-VAD (10 µM) in medium with an osmolality of 150 mOsm NaCl for 48 h. Cell viability was measured using the CCK-8 assay (*n* = 3). (**B**) Quantitative PCR was conducted to detect *Bcl-2*, *Bcl-xl*, and *Bax* mRNA expression. GAPDH expression was used as a loading control (*n* = 6). (**C**) Total cellular proteins were extracted from NFAT5 KO and WT cells. Protein levels of Bcl-2, Bcl-xL, and Bax were determined by Western blot analysis. The β-actin was used as a control for protein loading. (**D**–**F**). Quantifications of expression levels of Bax, Bcl-2, and Bcl-xl in NFAT5 KO and WT cells in normal medium and medium with an osmolality of 150 mOsm NaCl for 48 h (*n* = 3). Bars represent the mean ± S.E.M; n.s; non-significance vs. NM, NM, normal media; z-VAD, caspase inhibitor.

**Figure 6 ijms-22-06296-f006:**
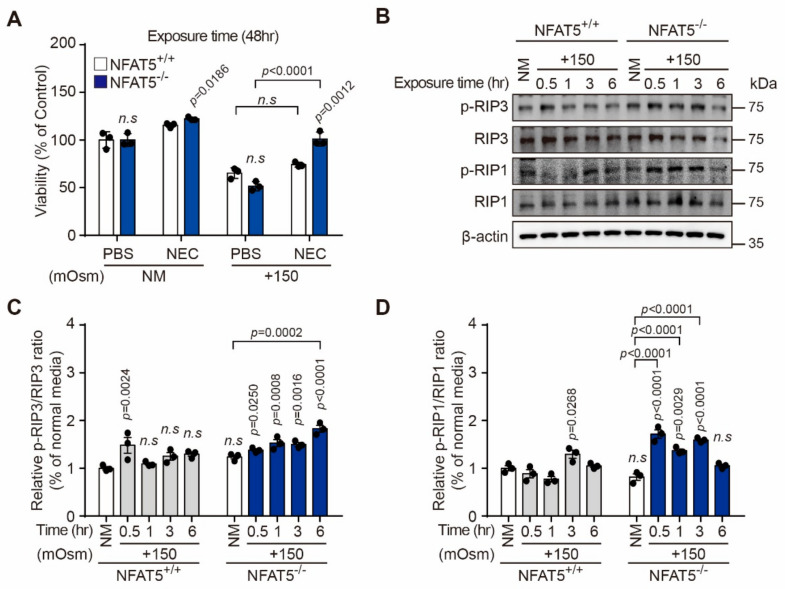
Effect of NFAT5 on activation of RIP1 and RIP3. (**A**) NFAT5 KO and WT cells were pretreated with NEC (10 µM) in NM and NM with 150 mOsm/L NaCl for 48 h. Cell viability was determined by the CCK-8 assay (*n* = 3). (**B**) Western blots evaluating expression of phosphorylated RIP1 (Ser166) and RIP3 (Ser227) and the corresponding total forms in NFAT5 KO and WT cells exposed to hyperosmotic stress. The β-actin was used as a loading control. (**C**,**D**) RIP1 and RIP3 protein concentrations were calculated by averaging the results obtained from three independent experiments (*n* = 3). Bars represent the mean ± S.E.M. n.s; non-significance vs. NM, Con, control; NEC, necrostatin-1; NM, normal media.

**Figure 7 ijms-22-06296-f007:**
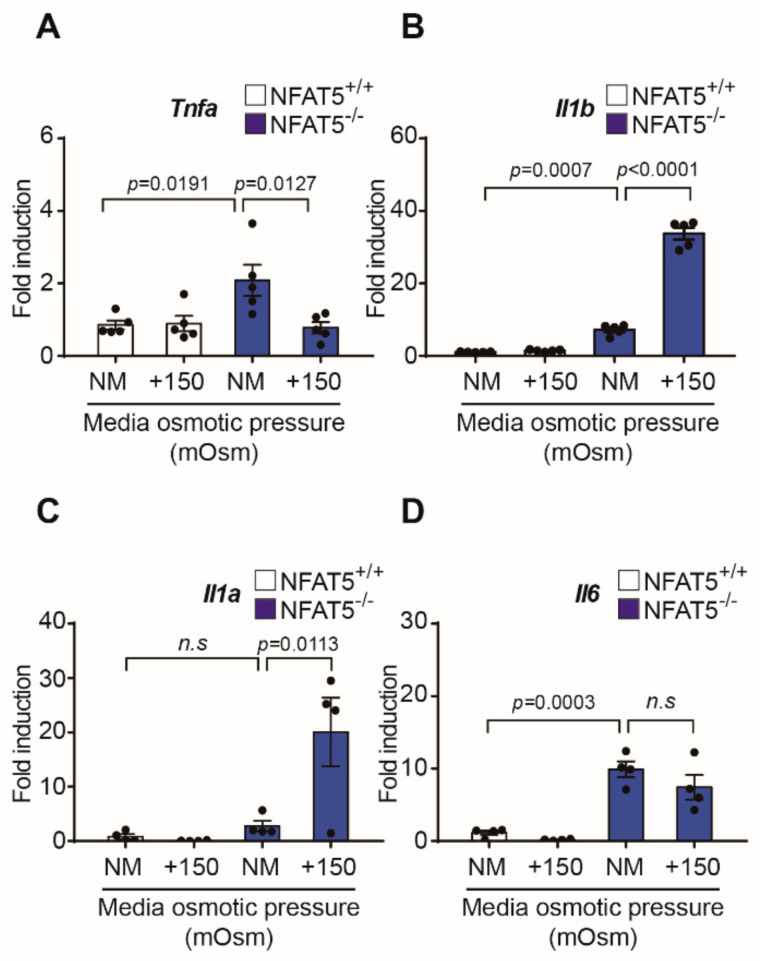
Changes in inflammatory cytokines to hypertonic conditions in NFAT5 WT and KO cell lines. (**A**–**D**) Effect of NFAT5 on changes of inflammatory cytokines (TNF-α, IL-1α, IL-1β, and IL-6) were investigated. NFAT5 KO and WT cells were treated in NM and NM with 150 mOsm/L NaCl for 24 h. Quantitative PCR was conducted to detect *Tnfα*, *Il1α*, *Il1β*, and *Il6* mRNA expression. GAPDH expression was used as a loading control (*n* = 6). Bars represent the mean ± S.E.M. n.s; non-significance vs. NM, NM, normal media.

**Table 1 ijms-22-06296-t001:** Sequences of the siRNA used for siRNA transfection with gene name, sequence.

	Gene	Primer	Sequences (5′–3′)
1	*Nfat5*	Sense	GACCAUGGUCCAAAUGCAA
		Antisense	UUGCAUUUGGACCAUGGUC
2	*Nfat5*	Sense	CUGGAUAACAGUCGGAUGU
		Antisense	ACAUCCGACUGUUAUCCAG

**Table 2 ijms-22-06296-t002:** Primer sequences used for mRNA expression analysis with gene name, sequence.

	Gene	Primer	Sequences (5′–3′)
1	*Bcl-2*	Forward	CAT GCG GCC TCT GTT TGA TT
		Reverse	TCACTTGTGGCCCAGATAGG
2	*Bcl-x1*	Forward	GGT CGC ATT GTG GCC TTT TT
		Reverse	CGT CGA TCC GAC TCA CCA AT
3	*Bax*	Forward	TGA GCA GAT CAT GAA GAC AGG G
		Reverse	TTG AGA CAC TCG CTC AGC TT
4	*tnf-α*	Forward	CCCTCACACTCAGATCATCTTCT
		Reverse	GCTACGACGTGGGCTACAG
5	*il-1α*	Forward	GCACCTTACACCTACCAGAGT
		Reverse	AAACTTCTGCCTGACGAGCTT
6	*il-1β*	Forward	GCTCGCCAGTGAAATGATGG
		Reverse	GTCCTGGAAGGAGCACTTCAT
7	*il-6*	Forward	TAGTCCTTCCTACCCCAATTTCC
		Reverse	TTGGTCCTTAGCCACTCCTTC
8	*Nfat1*	Forward	AGAATCCATCCTGCTGGTTC
		Reverse	TCCATGTAGCCATGGAGCTG
9	*Nfat2*	Forward	TCATTGACTGTGCCGGAATC
		Reverse	AAGTTGTGGCCAGACAGGAC
10	*Nfat3*	Forward	AGAACTGGACTCAGAGGATG
		Reverse	ATGGAGGTGATGCGGATG
11	*Nfat5*	Forward	TACCTCAGTCACCGACAGCAAG
		Reverse	CGACTGTTATCCAGCAAGTC

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
