# Peer review of "The Role of Nuclear Factor of Activated T Cells 5 in Hyperosmotic Stress-Exposed Human Lens Epithelial Cells"

_ijms, 2021, doi:10.3390/ijms22126296_

Round 1

Reviewer 1 Report

I have no more questions for this manuscript. 

Author Response

Thank you for your positive comments.

Reviewer 2 Report

The authors have evaluated the expression and function of NFAT5 in human lens epithelial cells induced by hyperosmotic stress. They found that  hyperosmolar stresses reduces HLEC viability and causes cell cycle arrest. They suggest that programmed cell death plays an important role in response to harmful stress.

Comments.

1.Some parts of the manuscript are written in red, is there any reason for this?

2. Which is the molecular mechanism by which hyperosmotic stress  increases  NFAT5 gene expression ? I suggest to investigate whetehr tarnscription factors induced by hyperosmotic stress may be involved in the modulation of gene expression.

3. The increase of NFAT5 protein expression is slightly increase in hyperosmotic respect to normal osmotic medium. Furthermore, the increase of NFAt5 protein expression is less evident in comparison to that found for  gene expression. Can you comment this?

3. What is the rationale for investigating the inflammatory cytokines in hyperosmotic exposure?

Author Response

Thank you for your detailed suggestions and comments.
We will reply to each of your comments as follows.

1.Some parts of the manuscript are written in red, is there any reason for this?

Responses) The parts marked in red in the manuscript are the authors' corrections to the comments pointed out by previous reviewers.

2. Which is the molecular mechanism by which hyperosmotic stress  increases  NFAT5 gene expression ? I suggest to investigate whetehr tarnscription factors induced by hyperosmotic stress may be involved in the modulation of gene expression.

Responses) Several papers revealed that hyperosmotic stress is involved in the regulation of NFAT5 gene expression. In this paper, reference 22 and 23 was used for explanation about relationship between hyperosmotic stress and NFAT5 gene expression. Moreover, this paper focused on the role of NFAT5 in hyperosmotic stress-exposed human lens epithelial cells rather than the regulatory mechanism of NFAT5.

3. The increase of NFAT5 protein expression is slightly increase in hyperosmotic respect to normal osmotic medium. Furthermore, the increase of NFAt5 protein expression is less evident in comparison to that found for  gene expression. Can you comment this?

Responses) The expressions of NFAT5 protein were significantly increased by hyperosmotic medium and these changes were statistically explained in figure and results. Furthermore, the increase pattern was similar to the expression pattern of mRNA. However, since posttranslational modifications affects to protein expression, there may be differences in the exact amount of expression compare with mRNA expression.

4. What is the rationale for investigating the inflammatory cytokines in hyperosmotic exposure?

Responses) Actually, the comments pointed out by previous reviewers are as follows:”Inflammatory responses are also important in the development of cataract. Does hyperosmotic stress induce inflammatory cytokines secretion in lens cells? It should be noted that NFAT5 is associated with several different inflammatory diseases and reactions.” Therefore, we measured changes in inflammatory cytokines by qPCR after 24 hours exposure to hypertonic condition in NFAT5 WT and KO cell lines, and then we added changes in inflammatory cytokines to the results by revision.

Round 2

Reviewer 2 Report

The paper is improved and the authors have answered to the comments adequately.

This manuscript is a resubmission of an earlier submission. The following is a list of the peer review reports and author responses from that submission.

Round 1

Reviewer 1 Report

The manuscript by Kim et al. studied the effect of hyper osmotic stress on human lens epithelial cells. The goal was to evaluate the molecular pathogenesis of cataract. They conclude that NFAT5 was associated with the hyperosmotic stress-induced death of lens cells. 

Comments

The title of this manuscript specifically stated the importance of NFAT5 in hyperosmotic stress-induced cell death. According to the data, NFAT5 was increased when the cells were treated with hyperosmotic stress, which suggested the importance of NFAT5 in the response to hyperosmotic stress. In NFAT5 knockout cells, the cell death was more serious than the WT cells treated with hyperosmotic stress. The results suggested a protective role of NFAT5 in hyperosmotic stress-induced cell death. The most confusing part is how the author jump into necroptosis? They only showed RIP-1 and 3 activation. But not how NFAT5 deletion increased RIP-1 and 3 activation. How does NFAT5, a transcription factor affect RIP-1 and 3 activation? In my opinion, it is the most important issue to answer.

Author Response

The title of this manuscript specifically stated the importance of NFAT5 in hyperosmotic stress-induced cell death. According to the data, NFAT5 was increased when the cells were treated with hyperosmotic stress, which suggested the importance of NFAT5 in the response to hyperosmotic stress. In NFAT5 knockout cells, the cell death was more serious than the WT cells treated with hyperosmotic stress. The results suggested a protective role of NFAT5 in hyperosmotic stress-induced cell death.

  1. The most confusing part is how the author jump into necroptosis?

Answer) As pointed out by the reviewer, there was a lack of logical comments in the necroptosis of human lens epithelial cells exposed hyperosmolar stress. I added the following comments in the first paragraph of revised manuscript:

“This current study was investigated about the cell death under hyperosmolar stress in HLECs. Exposure to hyperosmolar stresses reduces human lens epithelial cell viability and causes cell cycle arrest. Programmed cell death plays an important role in response to harmful stress and the regulation of homeostasis and various diseases. Apart from the classical apoptosis, recent basic experimental studies also discovered novel programmed cell death, necroptosis. This is the first programmed cell death that does not depend on the activation of caspases. Especially, RIP1, RIP3, and MLKL are the main molecules involved in necroptosis. In this study, the decrease in cell viability was not recovered when z-VAD, a caspase inhibitor that has a role in blocking apoptosis, was treated to NFAT5-WT and NFAT5-KO cells. On the other hand, when the RIP1 kinase inhibitor Nec-1 was treated, especially after hyperosmolar exposure in NFAT5-KO cells, it was confirmed that the decrease in cell viability was recovered. In addition, reductions in phopsporylated-RIP1 and -RIP3 also have been identified. Therefore, our results suggest that the osmo-adaptive response through NFAT5 enhances HLEC viability by suppressing necroptosis. To the best of our knowledge, this study is the first to describe the role of NFAT5 in HLECs upon hyperosmotic stress”

  1. They only showed RIP-1 and 3 activation. But not how NFAT5 deletion increased RIP-1 and 3 activation. How does NFAT5, a transcription factor affect RIP-1 and 3 activation? In my opinion, it is the most important issue to answer.

Answer) In fact, there is no experiment on how the depletion of NFAT5 causes necroptosis, that is, the mechanism by which the increase in phosphorylation of RIP1 and RIP3 occurs, and it is also limitation of this study. This mechanism study is another big project, and we are currently conducting follow-up research on this subject. As for the mechanism by which NFAT5 is involved in necroptosis, first, it may affect down signaling pathways such as NF-kB. And second, it may have influenced enzymes such as aldose reductase that are activated by NFAT5 to increase the intracellular osmolyte. Based on the reference thesis, the following contents has been added to the discussion:

“The major limitation of this study is that there is no mechanistic study on how depletion of NFAT5 is involved in necroptosis, that is, activation of RIP1 and RIP3. This mechanistic study of NFAT5 and necroptosis is a big project, and the authors are in progress as follow-up studies. The mechanism by which NFAT5 is involved in necroptosis is as follows. First, it acts on changes in the expression of down signaling pathways such as NF-kB. In other words, NFAT5 increases the activity of Nf-kB, and NFAT depletion suppresses the activation of Nf-kB and may be involved in necroptosis. Second, to increase intracellular osmolyte, NFAT5 affects the expression of enzymes such as aldose reductase, which are typically activated by NFAT5. It is thought that the depletion of NFAT5 reduces the acitivity of these aldose reductases, and as a result, necroptosis may be induced by several intermediate products.”

Reviewer 2 Report

This manuscript was largely revised from the previous version. The authors addressed all my concerns. This manuscript is ready to be published. 

Author Response

Thank you for your positive comments.

Round 2

Reviewer 1 Report

The author should perform the experiments that I requested. 

They only showed RIP-1 and 3 activation. But not how NFAT5 deletion increased RIP-1 and 3 activation. How does NFAT5, a transcription factor affect RIP-1 and 3 activation? In my opinion, it is the most important issue to answer.